# Can Perfusion-Based Brain Tissue Oxygenation MRI Support the Understanding of Cerebral Abscesses In Vivo?

**DOI:** 10.3390/diagnostics13213346

**Published:** 2023-10-30

**Authors:** Michael Knott, Philip Hoelter, Liam Soder, Sven Schlaffer, Sophia Hoffmanns, Roland Lang, Arnd Doerfler, Manuel Alexander Schmidt

**Affiliations:** 1Department of Neuroradiology, Friedrich-Alexander-Universität Erlangen-Nürnberg (FAU), Schwabachanlage 6, 91054 Erlangen, Germanymanuel.schmidt@uk-erlangen.de (M.A.S.); 2Department of Neurosurgery, Friedrich-Alexander-Universität Erlangen-Nürnberg (FAU), Schwabachanlage 6, 91054 Erlangen, Germany; 3Department of Microbiology, Friedrich-Alexander-Universität Erlangen-Nürnberg (FAU), Wasserturmstraße 3, 91054 Erlangen, Germany

**Keywords:** brain abscess, DSC-PWI, brain tissue oxygenation

## Abstract

Purpose: The clinical condition of a brain abscess is a potentially life-threatening disease. The combination of MRI-based imaging, surgical therapy and microbiological analysis is critical for the treatment and convalescence of the individual patient. The aim of this study was to evaluate brain tissue oxygenation measured with dynamic susceptibility contrast perfusion weighted imaging (DSC-PWI) in patients with brain abscess and its potential benefit for a better understanding of the environment in and around brain abscesses. Methods: Using a local database, 34 patients (with 45 abscesses) with brain abscesses treated between January 2013 and March 2021 were retrospectively included in this study. DSC-PWI imaging and microbiological work-up were key inclusion criteria. These data were analysed regarding a correlation between DSC-PWI and microbiological result by quantifying brain tissue oxygenation in the abscess itself, the abscess capsula and the surrounding oedema and by using six different parameters (CBF, CBV, CMRO2, COV, CTH and OEF). Results: Relative cerebral blood flow (0.335 [0.18–0.613] vs. 0.81 [0.49–1.08], *p* = 0.015), relative cerebral blood volume (0.44 [0.203–0.72] vs. 0.87 [0.67–1.2], *p* = 0.018) and regional cerebral metabolic rate for oxygen (0.37 [0.208–0.695] vs. 0.82 [0.55–1.19], *p* = 0.022) were significantly lower in the oedema around abscesses without microbiological evidence of a specific bacteria in comparison with microbiological positive lesions. Conclusions: The results of this study indicate a relationship between brain tissue oxygenation status in DSC-PWI and microbiological/inflammatory status. These results may help to better understand the in vivo environment of brain abscesses and support future therapeutic decisions.

## 1. Introduction

Despite continuous development in medical diagnostics and therapy, the clinical condition of a brain abscess is still a life-threatening disease with significant long-term morbidity, prolonged convalescence and a mortality rate of up to 20% [1,2,3,4]. 

Diagnosis of an abscess of the brain parenchyma is mainly based on MRI followed by drainage, evacuation and microbiological analysis of the abscess itself. The analysis of the hereby-acquired samples regarding the specific microbiological spectrum which caused the infection is important for a calculated drug therapy [5,6]. Nevertheless, there are up to 28% of cases in which microbiological analysis remains negative [7].

Recommended sequences for MRI in a possible case of an intracranial abscess are diffusion weighted imaging (DWI) with apparent diffusion coefficient maps (ADC), T2w-/FLAIR- sequences, as well as native and contrast-enhanced T1- sequences to identify the typical core, capsule and surrounding oedema of an abscess in a manner that is as safe as possible [6,8,9]. At the moment, the diagnostic potential of innovative MRI techniques such as dynamic susceptibility contrast perfusion weighted imaging (DSC-PWI) has only a limited impact in the diagnostic procedure of cerebral abscesses and is currently mostly limited to challenging cases for concretising differential diagnosis [10,11]. 

Emerging techniques for brain tissue oxygenation and microvascular analysis based on DSC perfusion have already shown its diagnostic potential in the setting of ischemic and neoplastic disorders and might also have potential towards expanding the role of MRI in the diagnostic of infectious brain disorders. Currently, DSC perfusion is an important tool in stroke diagnostics regarding the penumbra as well as in tumour diagnostics regarding the interpretation of progress vs. pseudo-progress in patients undergoing therapy.

The new parameters of brain tissue oxygenation analysis allow a view into angiogenesis (Cerebral blood flow (CBF)), oxygenation status (Cerebral blood volume (CBV) and Cerebral metabolic rate of oxygen (CMRO2)), oxygen utilization by brain tissue (Oxygen extraction fraction (OEF)) and distribution of microvascular flow patterns (Capillary transit time heterogeneity (CTH)). These new diagnostic parameters are extending the diagnostic understanding of the metabolic environment in and around pathologic brain processes [12,13,14,15,16,17,18,19].

Our research evaluates the potential of DSC-PWI-based innovative brain tissue oxygenation analysis regarding the in vivo environment of brain abscesses with special focus on the microbiological and inflammatory status.

## 2. Material and Methods

### 2.1. Patients

For this study, the institutional database of the department of neuroradiology of the university hospital Erlangen was retrospectively screened for patients with intraparenchymal abscesses of the brain. All clinical data and imaging were initially obtained for clinical purposes.

Between January 2013 and March 2021, a total of 113 patients were identified with a first-time diagnosis of this disease and MRI-imaging prior to calculated therapy.

Inclusion criteria for our analysis were the following: (1) presence of perfusion imaging in initial MRI, (2) imaging quality sufficient for reliable analysis, (3) abscess configuration suitable for analysis (volume of the abscess > 0.5 mL, thickness of the capsula > 2 mm) and (4) presence of complete medical work-up.

Finally, 34 patients with a total of 45 abscesses were included in our analysis as shown in Figure 1.

### 2.2. Imaging

MR imaging was performed using a 1.5 T Scanner (Siemens Magnetom Aera, Siemens Healthineers AG, Erlangen, Germany). The examination protocol included a DWI sequence (including ADC maps), a fluid-attenuated inversion recover (FLAIR) sequence, DSC perfusion with leakage correction and an isotropic contrast-enhanced magnetization-prepared rapid gradient echo (MP-RAGE) T1 sequence (Table 1).

DSC perfusion imaging was conducted by intravenous administration of 0.5 mmol/mL/kg Dotarem (Guerbet, Villepinte, France) or Dotagraf (Jenapharm GmbH & Co. KG, Jena, Germany) at a flowrate of 3.5 mL/s, followed by a 20 mL saline flush at the same flowrate. IV injection was started at the fourth time point of DSC perfusion via a 20 Gauge peripheral venous catheter placed in a cubital vein.

### 2.3. Postprocessing

Postprocessing of the DSC-PWI datasets was performed in a standardized manner using a commercially available post-processing software (Cercare Medical Neurosuite, Cercare Medical, Aarhus, Denmark). This software calculated relative cerebral blood volume, relative cerebral blood flow, cerebral metabolic rate of oxygen, coefficient of variance, capillary transit time heterogeneity and oxygen extraction fraction (Table 2).

Further postprocessing was performed using the MR neurology workflow of syngo.via (Siemens Healthineers AG, Erlangen, Germany). In this workflow, automatic co-registration of the morphologic sequences and the calculated DSC-PWI maps was performed to allow and guarantee simultaneous and corresponding region-of-interest (ROI) placement in all maps. Regions of interest were (1) the centre of the abscess shown by the restriction in DWI, (2) the capsule of the abscess represented by the enhanced ring on post-contrast MP-RAGE and (3) the surrounding oedema depictured an FLAIR. To set the acquired results in correlation with the healthy contralateral parenchyma, normalization was performed by automatically mirroring the placed ROIs (Figure 2).

Manual adjustment was performed in cases where ROIs were placed in non-representative locations (e.g., subarachnoid space, vessels or bone). Furthermore, volumetric analyses of the abscess core and the oedema were conducted on DWI, respectively, and the FLAIR sequence. ROI placement and volumetric analyses were performed manually, in consent, by two physicians (MK and PH).

Volumetric analysis was performed manually by defining the affected area on each slice and adjusting this result with the thickness of the slice. For further analysis in patients with multiple abscesses, the volumes of the single lesions regarding core (DWI) and oedema (FLAIR) were added, whereas the arithmetic mean was taken for the individual results of DSC-PWI parameters.

### 2.4. Diagnostic Microbiology

Samples were collected by neurosurgical drainage, abscess evacuation or, in case of meningitis symptoms, through lumbar puncture to obtain cerebrospinal fluid (CSF). All samples were analyzed by microscopy after Gram-staining and by aerobic and anaerobic culture for 7 days. Bacterial or fungal colonies were identified with MALDI-TOF mass spectrometry (Bruker Daltonik GmbH, Bremen, Germany) as described before [24]. In many cases, molecular detection of bacteria in the samples was performed by PCR amplification of the 16S rDNA gene and, if positive, conventional Sanger sequencing of amplicons as previously described [25].

### 2.5. Statistical Analysis

Statistical analysis was performed using commercially available software (IBM^®^ SPSS^®^ Statistics version 19, Chicago, IL, USA). Normal distribution was tested using the Shapiro–Wilk test. In case of normal distribution, data are presented as mean ± standard deviation. Non-normally distributed data are presented as median with interquartile range. The Mann–Whitney U test was applied to assess for statistical significance. For quality evaluation of the results, receiver operating characteristic (ROC) curves were calculated. Statistical significance was considered for a *p* value of less than 0.05.

## 3. Results

A total of 34 patients were included in this study (41.2% female). Mean age was 56.7 (±14.9) years. Median number of abscesses in an individual patient was one (1-1), maximal number of abscesses in a single patient was seven. In seven patients (20.6%, eight abscesses) out of these 34 cases, no microbiological detection of specific pathogens was possible, whereas evidence of microbiological pathogens was obtained in 27 cases (79.4%, 37 abscesses). Microbiological work-up was performed in 15 cases (41.1%) by microbiological culture and in 19 cases (55.9%) by culture and PCR analysis. 

For further evaluation, the two collectives with positive and negative microbiological result were compared regarding clinical items (CRP, leucocyte count) and MRI imaging (volume of core/oedema, ADC value, DWS-PWI), with special focus on brain oxygenation DCS-PWI results. Full overview of all test items is given in Table 3.

For clinical items, testing showed mean CRP values of 2.1 [0.9–5.1] vs. 28.3 [10.4–71.4] (*p* = 0.003) and a mean count of leucocytes of 6.89 [5.43–12.16] vs. 10.69 [8.7–14.96] (*p* = 0.025), comparing microbiologically negative and postitive patients. 

In morphological MR imaging analysis, the volume of the abscess cavity was 3.927 mL [1.436–34.684] vs. 7.478 mL [2.499–27.36] (*p* = 0.677) and that of the oedema was 26.643 mL [12.761–77.427] vs. 45.705 mL [20.319–91.73] (*p* = 0.427). The ADC value in the abscess cavity was 0.766 [0.544–0.863] vs. 0.732 [0.585–0.892] (*p* = 0.949).

Statistically significant differences could be demonstrated for the brain oxygenation DSC-PWI parameter between the microbiological negative and positive groups for the values of rCBF (0.335 [0.18–0.613] vs. 0.81 [0.49–1.08], *p* = 0.015), rCBV (0.44 [0.203–0.72] vs. 0.87 [0.67–1.2], p = 0.018) and rCMRO2 (0.37 [0.208–0.695] vs. 0.82 [0.55–1.19], *p* = 0.022) regarding the oedema.

Further characteristics of the two collectives are illustrated in Table 3. A summary of the detected microbiological results and the detected inflammatory focus is appended in Table 4 and Table 5.

When performing ROC curve analysis, AUC was quite similar for rCBF (0.821 (0.619–1); *p* = 0.015), rCBV (0.812 (0.602–1); *p* = 0.018) and rCMRO2 (0.802 (0.598–1); *p* = 0.022) (Figure 3).

In total, six patients had more than one lesion. One patient with two separate abscesses and microbiological analysis of both lesions had no result in microbiological work-up. Five patients with more than one abscess had evidence of a microbiological flora in the abscesses. Four of these five patients had each two separate lesions with neurosurgical drainage and microbiological work-up of both lesions. One of these five cases had eight abscesses distributed across the whole brain. Seven of these lesions fulfilled the inclusion criteria of this study. Two of these seven lesions were drained and analysed. These 2 lesions were the biggest of the lesions (core volume 10.15 mL and 16.53 mL), each with a large oedema (49.45 mL and 24.6 mL). One of these lesions also had the highest values of rCMRO2 (1.33), rCBV (1.49) and rCBF (1.32). Further details regarding this single case analysis are depicted in Table 6.

## 4. Discussion

To our knowledge, this is the first study to analyse brain tissue oxygenation measured with DSC-PWI in abscessing parenchymal infections of the brain in the microbiological context.

In our cohort and regarding the oedema, the values of rCBF, rCBV and rCMRO2 were significantly lower in patients without microbiological evidence of a specific pathogen in comparison to the collective with positive results in microbiological analysis.

DSC-PWI in general and the current research regarding brain tissue oxygenation measured with DSC-PWI in stroke and tumour microvascularity demonstrate significant advances for the diagnostic in ischemic and neoplastic brain disease. Taking this research into account, our results imply a difference in microvascular architecture and metabolism in the oedema of abscesses in dependency of the microbiological flora and the inflammatory status in analogy to the topic of progress vs. therapy-related pseudoprogress in astrocytomas [16,21,22,23]. Our results regarding the perfusion of the contrast-enhancing capsule—also not showing statistically significant differences between our subgroups—are in consistence with the results of capsula perfusion in the works of Erdogan et al. and Muccio et al. [10,11].

CBF as a parameter of angiogenetic status as well as CBV and rCMRO2 as parameters of oxygenation status showed significant differences between the subpopulations of our collective. 

This result can be interpreted as a lower rate of oxygen demand and therefore of metabolic activity as well as a lower rate of angiogenetic dependency in microbiological negative patients. A more consolidated infection does not need as much perfusion in terms of vascular dependence and accessibility as well as actual oxygen supply in comparison to a stadium of more active immune response which has a higher vascular and oxygenation demand for better accessibility and a higher rate of immunological activity [12,17,26].

In addition, the values of CRP and leucocyte samples taken in timely correlation to the MRI examination showed a tendency toward lower counts for both parameters in the microbiologically negative subgroup. 

It is widely documented that systemic inflammatory parameters such as the CRP value and leucocyte count are not fully trustworthy in cases of brain abscesses [2,4,27].

Nevertheless, the lower results in DSC perfusion in correlation with the lower values of CRP and leucocyte count may imply that immune response in abscesses without microbiological result is either in an earlier or more developed/consolidated stage of inflammatory response in which the total number of pathogens in the abscesses cavity is beyond the level of detection, whereas in patients with a relatively higher perfusion in the oedema, the higher values of systemic immune parameters may indicate a stage of a more active immune response, in which the total count of pathogens is higher and therefore, in microbiological analysis, more likely to be detected.

These results were further supported by a single case analysis in a patient of our study population with multiple separate abscesses. As neurosurgical drainage of all lesions was not possible; only the largest lesions in non-eloquent brain regions were targeted for surgery. Microbiological work-up of the acquired samples demonstrated specific bacteria as abscess flora. Retrospectively, the comparison of DSC-PWI values of each lesion supported the neurosurgical choice as the targets include the lesion with the highest DSC-PWI values, and therefore—following our results—the lesion with the highest possibility for a successful microbiological work-up.

Our study has five major limitations, the first one being the limited number of patients in total and especially in the subgroup without microbiological evidence. This results from intraparenchymal abscesses being a rare disease and DSC-PWI not being an established tool in the standard MRI of brain abscesses.

The second limitation is the study design. As a retrospective and cross-sectional analysis, it misses the power of a prospective and longitudinal study design with repeating MRI imaging (including DSC-PWI) and maybe even repeated microbiological testing.

The third limitation lies in the inhomogeneity of the medical history of our patients, the sample material and the microbiological analysis. We accepted cases with comorbidities and past medical history, which may have influenced the development of an abscess (in one case, even with meningitis 2 months earlier and with full recovery having occurred in the meantime). We also accepted microbiological results regardless of the nature of the sample and accepted results from microbiological culturing as well as from PCR testing. We did this based on the retrospective nature of our research and the real-world data it represents.

The fourth bias may lie in the used software for DSC perfusion analysis. We used this software as it incorporates not only established parameters such as CBF and CBV, but also new brain oxygenation parameters as mentioned above. As the mathematic algorithm for calculating DSC-PWI maps differs between different analysing software, this may also influence the transferability of our results.

The final limitation is the manual placement of the ROIs. This was performed in consent by two experienced neuroradiologists in representative locations to reduce this bias. Nonetheless, the homogeneity of perfusion in and around a brain abscess is questionable and needs further research.

In conclusion, the results of this study indicate a relationship between perfusion status, brain tissue oxygenation and microbiological/inflammatory status. This relationship still needs further analysis but may support the future diagnostic and therapy work-up in patients with brain abscesses (e.g., in the form of guided surgery).

## Figures and Tables

**Figure 1 diagnostics-13-03346-f001:**
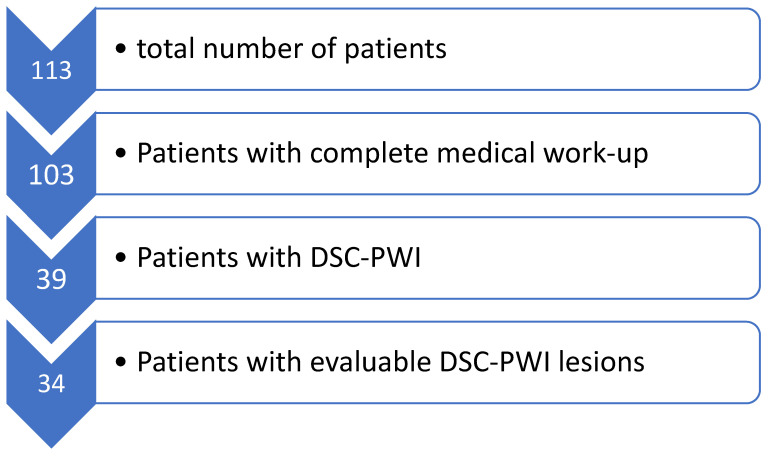
Flow chart of inclusion criteria of this study; DSC-PWI = dynamic susceptibility contrast perfusion weighted imaging.

**Figure 2 diagnostics-13-03346-f002:**
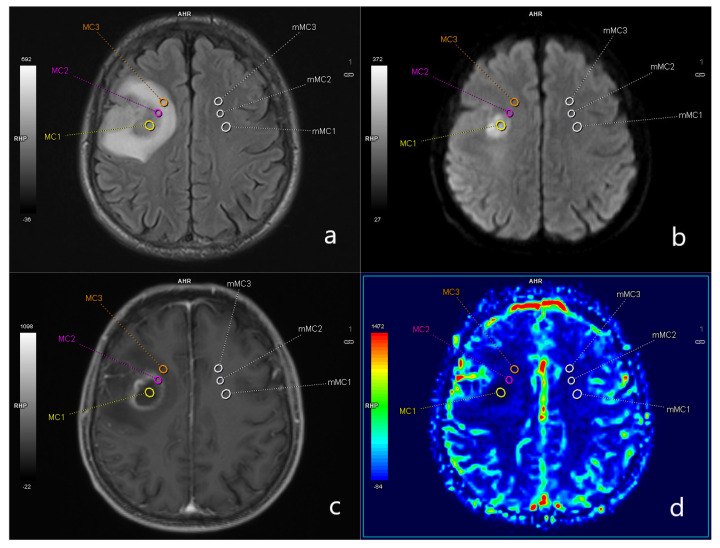
ROI placement in a representative case. ROI placement in a representative case on (**a**) FLAIR, (**b**) DWI, (**c**) contrast-enhanced T1–MPRAGE and (**d**) DSC–PWI maps in the abscess cavity (MC 1, yellow ROI), the capsula UMC 2, violet ROI) and the oedema (MC 3, orange ROI). Also shown are mirrored ROIs for each location (mMC 1–3) in the contralateral hemisphere for correlation with healthy brain parenchyma.

**Figure 3 diagnostics-13-03346-f003:**
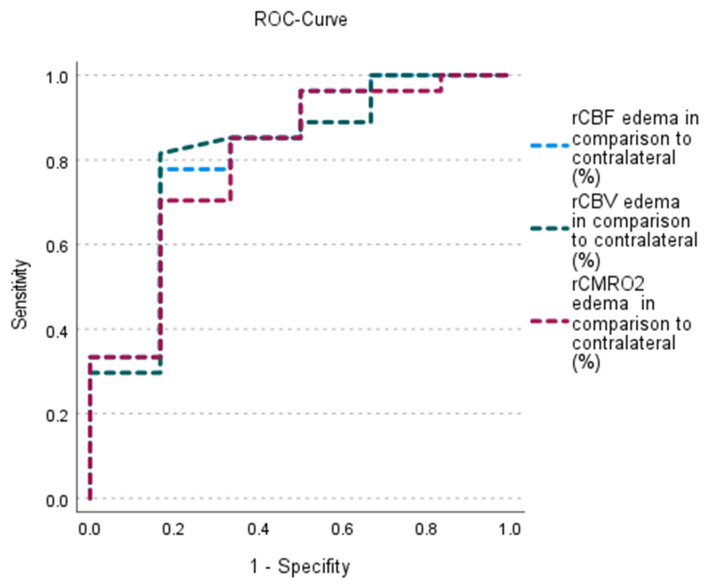
ROC curve analysis. Receiver Operating Characteristic curve for rCBF, rCBV and rCMRO2 to visualise diagnostic potential/power.

**Table 1 diagnostics-13-03346-t001:** Selected MRI sequence parameters.

	FLAIR	DWI	DSC Perfusion with Leakage Correction	Contrast-Enhanced T1 MP-RAGE
TR (ms)	9000	7600	2010	2200
TE (ms)	110	86	30	2.67
Flip angle (°)	150	90	90	8
FOV (mm^2^)	230	230	230	250
Matrix (pixel)	256 × 256	324 × 372	128 × 128	256 × 256
Voxel size (mm)	0.9 × 0.9 × 5	1.2 × 1.2 × 5	1.8 × 1.8 × 3	1.0 × 1.0 × 1.0
Acquisition time (min)	2:26	1:25	1:48	4:59

TR = repetition time; TE = echo time; FOV = field of view; FLAIR = fluid-attenuated inversion recovery; DWI = diffusion-weighted imaging; DSC = dynamic susceptibility contrast, MP-RAGE = magnetization-prepared rapid gradient echo.

**Table 2 diagnostics-13-03346-t002:** Perfusion parameters.

Parameter	Description	Acquisition
Cerebral blood flow (CBF)	Reflecting the angiogenesis	Software-based measurement
Cerebral blood volume (CBV)	Reflecting the oxygenation/microvascular status	Software-based measurement
Cerebral metabolic rate of oxygen (CMRO2)	Reflecting the oxygenation/microvascular status	Calculated using OEF and CBF
Coefficient of variance (COV)	Reflecting relative transit-time heterogeneity	Calculated using CTH, CBF and CBV
Capillary transit time heterogeneity (CTH)	Reflecting the distribution of microvascular flow patterns	Software-based measurement
Oxygen extraction fraction (OEF)	Reflecting the efficiency of oxygen utilization by the tissue	Calculated using CBV

Based on [12,13,14,15,16,17,18,19,20,21,22,23].

**Table 3 diagnostics-13-03346-t003:** Characteristics.

	Patients without Microbiological Result (7 Patients; 8 Abscesses)	Patients with Microbiological Result (27 Patients; 37 Abscesses)	*p*-Value
Age in years	59.43 (± 19.865)	56.15 (± 14.912)	0.456
Gender			
- Female	1 (14.3%)	15 (55.6%)	0.055
- Male	6 (85.7%)	12 (44.4%)	
Number of abscesses	1 [1-1]//6 × 1, 1 × 2	1 [1-1]//21 × 1, 3 × 2, 1 × 4, 1 × 7	0.748
Side			
- Rights	4 (50%)	17 (45.9%)	
- Left	3 (37.5%)	18 (48.6%)	
- Infratentorial	1 (12.5%)	2 (5.4%)	
Location			
- Frontal	3 (37.5%)	14 (37.8%)	
- Parietal	1 (12.5%)	7 (18.9%)	
- Temporal	2 (25%)	7 (18.9%)	
- Occipital	1 (12.5%)	6 (16.2%)	
- Infratentorial	1 (12.5%)	2 (5.4%)	
- Basal ganglia	0	1 (2.7%)	
CRP (mg/dl)	2.1 [0.9–5.1]	28.3 [10.4–71.4]	0.003
Pathologic CRP (>5mg/dL)	2 (28.6%)	23 (85.2%)	
Leucocyte count (×10³/µL)	6.89 [5.43–12.16]	10.69 [8.7–14.96]	0.025
Leucocytes in relation to reverence			
- Leukopenia (<4 × 10³/µL)	1 (14.3%)	0	
- Leukozytosis (>11.5 × 10³/µL)	2 (28.6%)	12 (44.4%)	
Additional PCR testing for microbiological analysis	4 (57.1%)	15 (55.6%)	
Volume of abscess (in mL)	3.927 [1.463–34.684]	7.487 [2.499–27.36]	0.677
DWI *	2.84 [2.05–4.59]	2.85 [2.42–3.42]	0.782
ADC *			
▪ Abscess	0.766 [0.544–0.863]	0.732 [0.585–0.892]	0.949
▪ Capsula	1.153 [0.72–1.652]	1.171 [1.02–1.652]	0.209
▪ Oedema	1.689 [1.229–2.24]	1.643 [1.457–2.026]	0.779
Volume of oedema (in ml)	26.643 [12.761–77.427]	45.705 [20.319–91.73]	0.427
Carecare–COV [CMN] *			
▪ Abscess	0.108 [0.71–1.16]	1.12 [1.04–1.27]	0.268
▪ Capsula	0.68 [0.56–0.8]	0.8 [0.71–0.9]	0.153
▪ Oedema	0.935 [0.89–1.11]	0.93 [0.89–1.02]	0.64
Carecare–rCMRO2 *			
▪ Abscess	0.3 [0.23–0.44]	0.21 [0.12–0.34]	0.269
▪ Capsula	0.71 [0.67–1.13]	0.84 [0.57–1.22]	0.609
▪ Oedema	0.37 [0.208–0.695]	0.82 [0.55–1.19]	0.022
Carecare–OEF [CMN] *			
▪ Abscess	1.17 [0.92–1.48]	1.14 [0.95–1.37]	0.686
▪ Capsula	1.39 [1.05–1.52]	1.31 [1.07–1.56]	0.881
▪ Oedema	1.17 [1.04–1.615]	1.1 [0.91–1.19]	0.225
Carecare–CTH [CMN] *			
▪ Abscess	1.54 [0.81–1.92]	1.78 [1.38–2.75]	0.154
▪ Capsula	0.94 [0.9–1.05]	1.16 [0.84–1.47]	0.154
▪ Oedema	1.155 [1–2.113]	1.03 [0.67–1.37]	0.262
Carecare–rCBV [CMN] *			
▪ Abscess	0.36 [0.33–0.42]	0.29 [1.5–0.54]	0.406
▪ Capsula	0.87 [0.67–1.21]	1.01 [0.73–1.54]	0.58
▪ Oedema	0.44 [0.203–0.72]	0.87 [0.67–1.2]	0.018
Carecare–rCBF [CMN] *			
▪ Abscess	0.24 [0.19–0.39]	0.18 [0.08–0.43]	0.406
▪ Capsula	0.57 [0.44–0.77]	0.68 [0.37–0.93]	0.848
▪ Oedema	0.335 [0.18–0.613]	0.81 [0.49–1.08]	0.015

* = in comparison to contralateral normal brain parenchyma; CRP = C-reactive protein, DWI = diffusion-weighted imaging, ADC = apparent diffusion coefficient, COV = coefficient of variance, rCMRO2 = cerebral metabolic rate of oxygen, OEF = oxygen extraction fraction, CTH = capillary transit time heterogeneity, rCBV = cerebral blood volume, rCBF = cerebral blood flow.

**Table 4 diagnostics-13-03346-t004:** Microbiological result.

Microbiological Result	Frequency
No microbiological result	7
*Streptococcus intermedius*	11
*Staphylococcus aureus*	4
*Fusobacterium nucleatum*	4
*Streptococcus constellatus*	3
*Agregatibacter aphrophilus*	2
*Streptococcus pneumonia*	1
*Nocardia farcinica*	1
*Prevotella oris*	1
*Cladophialophora bantiana*	1
*Parvimonas micra*	1
*Actinomyces meyeri*	1
*Propionibacterium acnes*	1
*Finegoldia magna*	1
*Enterococcus faecalis*	1
*Peptostreptococcus*	1
Gram negative rods *	1
Mixed flora *	2

In cases of a mixed bacterial flora of an abscess, bacteria were counted individually (* = not otherwise specified).

**Table 5 diagnostics-13-03346-t005:** Focus of primary infection.

Inflammatory Focus	Frequency	Percent
No focus	17	50.0
Iatrogenic	2	5.9
Immunocompromised	7	20.6
ENT	5	14.7
Cardiopulmonary	2	5.9
Septic condition	1	2.9

**Table 6 diagnostics-13-03346-t006:** Single case analysis in a patient with multiple abscesses.

	Side	Location	Core Volume (in mL)	Oedema Volume (in mL)	Capsula Thickness (in mm)	Oedema rCMRO2 ^‡^	Oedema rCBV (CMN) ^‡^	Oedema rCBF (CMN) ^‡^
Abscess 1 *	Right	Parietal	10.15	49.45	3	1.33	1.49	1.32
Abscess 2	Right	Temporal	7.83	13.31	2	1.08	1.19	0.97
Abscess 3 *	Right	Occipital	16.53	24.6	4	0.62	0.74	0.58
Abscess 4	Right	Frontal	4.42	30.74	4	1.26	1.18	1.08
Abscess 5	Left	Frontal	0.5	0.68	2	1.4	1.38	1.4
Abscess 6	Right	Frontal	1.26	0.76	3	0.25	0.35	0.16
Abscess 7	Right	Temporal	0.86	1.18	3	1.95	1.43	2.01

* = drained and microbiological analysed lesions, ^‡^ = in comparison to contralateral normal brain parenchyma.

## Data Availability

The datasets generated and analysed during the current study are available from the corresponding author on reasonable request.

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
