# Peer review of "Can Perfusion-Based Brain Tissue Oxygenation MRI Support the Understanding of Cerebral Abscesses In Vivo?"

_diagnostics, 2023, doi:10.3390/diagnostics13213346_

Round 1

Reviewer 1 Report

Comments and Suggestions for Authors

1) “Emerging techniques for brain tissue oxygenation and microvascular analysis based on DSC-perfusion have already shown its diagnostic potential in the setting of ischemic and neoplastic disorders and might also have potential towards expanding the role of MRI in the diagnostic of infectious brain disorders.”

It would significantly strengthen the introduction of this paper to elaborate on both currently-available and emerging techniques in more detail within the introduction. Although brain abscesses are a condition of significant and health importance, this is not necessarily remarked upon in the paper.

2) Formatting and capitalization in Table 4 should be consistent among rows and columns.

3) Formatting should be more consistent among all tables.

4) “Picture 1” should be re-listed as Figure 2, and numbering of figures updated to reflect this.

5) The authors state in the Discussion section that this study is "the first study to analyse brain tissue oxygenation measured with DSC-PWI in abscessing parenchymal infections of the brain in the microbiological context" however, there appears to be little description of the importance of this technique overall as a diagnostic tool. Both the introduction and discussion sections could be strengthened to support the authors' work and the importance of this technique in practice. 

Comments on the Quality of English Language

1) Figure 1. Capitalization should be consistent among text elements.

2) The acronym ROI should be defined at the first use.

3) In the results section it is stated that “ABC value in the abscess cavity was 0.766 [0.544 – 0.863] vs. 0.732 [0.585 – 0.892] (p = 0.949).” Previously the value was referred to as the ADC value. Please correct and define the acronym.

4) “Considering the principals of DSC parameters in general and taking into account the research on stroke and tumour microvascularity, our results implicate a difference in microvascular architecture and metabolism in the edema of abscesses in dependency of the microbiological flora and the inflammatory status in analogy to the topic of progress vs. therapy-related pseudoprogress in astrocytomas [16, 21 – 23]”

This sentence is long and unwieldy, it may benefit the authors to elaborate more effectively in shorter sentences that more directly make their point.

Author Response

Dear reviewer,

Thank you very much for your dedicated analysis and kind review of our work.

We addressed your recommendations in the 'Comments and Suggestions for Authors' section as follows:

1) “Emerging techniques for brain tissue oxygenation and microvascular analysis based on DSC-perfusion have already shown its diagnostic potential in the setting of ischemic and neoplastic disorders and might also have potential towards expanding the role of MRI in the diagnostic of infectious brain disorders.”

It would significantly strengthen the introduction of this paper to elaborate on both currently-available and emerging techniques in more detail within the introduction. Although brain abscesses are a condition of significant and health importance, this is not necessarily remarked upon in the paper.

To 1) we strengthened the mentioned points by adding:

- Page 5: 'significant long-term morbidity, prolonged convalescence and'

- Page 5: 'Currently DSC-perfusion is an important tool in stroke diagnostics regarding the penumbra as well as in tumour diagnostics regarding the interpretation of progress vs. pseudo-progress in patients undergoing therapy.'

- Page 6: 'These new diagnostic parameters are going to extending the diagnostic understanding of the metabolic environment in and around pathologic brain processes'

2) Formatting and capitalization in Table 4 should be consistent among rows and columns.

3) Formatting should be more consistent among all tables.

To 2) and 3) we standardized the layout of all tables

4) “Picture 1” should be re-listed as Figure 2, and numbering of figures updated to reflect this.

To 4) we corrected the nomenclature of our figures and changed Picture 1 to figure 2.

5) The authors state in the Discussion section that this study is "the first study to analyse brain tissue oxygenation measured with DSC-PWI in abscessing parenchymal infections of the brain in the microbiological context" however, there appears to be little description of the importance of this technique overall as a diagnostic tool. Both the introduction and discussion sections could be strengthened to support the authors' work and the importance of this technique in practice. 

To 5) in the introduction we strengthened the importance of DSC-PWI by adding:

- Page 5: 'Currently DSC-perfusion is an important tool in stroke diagnostics regarding the penumbra as well as in tumour diagnostics regarding the interpretation of progress vs. pseudo-progress in patients undergoing therapy.'

- Page 6: 'These new diagnostic parameters are going to extending the diagnostic understanding of the metabolic environment in and around pathologic brain processes'

In the discussion section, we strengthened the importance of DSC-PWI by adding:

  • Page 19: 'DSC-PWI in general and the currant research regarding brain tissue oxygenation measured with DSC-PWI in stroke and tumour microvascularity demonstrate significant advances for the diagnostic in ischemic and neoplastic brain diesease.'

We addressed your recommendations in the 'Comments on the Quality of English Language' section as follows:

  • Figure 1. Capitalization should be consistent among text elements.

To 1) Capitation has been equalized

  • The acronym ROI should be defined at the first use.

To 2) ROI has been defined at the first use

  • In the results section it is stated that “ABC value in the abscess cavity was 0.766 [0.544 – 0.863] vs. 0.732 [0.585 – 0.892] (p = 0.949).” Previously the value was referred to as the ADC value. Please correct and define the acronym.

To 3) ADC was define and the mentioned section was corrected

  • “Considering the principals of DSC parameters in general and taking into account the research on stroke and tumour microvascularity, our results implicate a difference in microvascular architecture and metabolism in the edema of abscesses in dependency of the microbiological flora and the inflammatory status in analogy to the topic of progress vs. therapy-related pseudoprogress in astrocytomas [16, 21 – 23]”

This sentence is long and unwieldy, it may benefit the authors to elaborate more effectively in shorter sentences that more directly make their point.

To 4) We tried to clarify our point by recreating the mentioned sentence:

'DSC-PWI in general and the currant research regarding brain tissue oxygenation measured with DSC-PWI in stroke and tumour microvascularity demonstrate significant advances for the diagnostic in ischemic and neoplastic brain diesease. Taking this research into account, our results implicate a difference in microvascular architecture and metabolism in the edema of abscesses in dependency of the microbiological flora and the inflammatory status in analogy to the topic of progress vs. therapy-related pseudoprogress in astrocytomas [16, 21 – 23].'

Thank you very much for your kind review.

Reviewer 2 Report

Comments and Suggestions for Authors

Aim of this study was to evaluate brain tissue oxygenation measured with dynamic susceptibility contrast perfusion- weighted imaging (DSC-PWI) in patients with brain abscess and its potential benefit for a better understanding of the environment in and around brain abscesses. Using a local database 34 patients (with 45 abscesses) with brain abscesses were retrospectively included in this study. DSC-PWI imaging and microbiological work-up were key inclusion criteria. This data was analysed regarding a correlation between DSC-PWI and microbiological result by quantifying brain tissue oxygenation in the abscess itself, the abscess capsula and the surrounding edema and by using six different parameters. The results of this study indicate a relationship between brain tissue oxygenation status in DSC-PWI and microbiological/inflammatory status. These results may help to better understanding the in-vivo environment of brain abscesses and support future therapeutic decisions.

Excellent clinical study. No of recruited patients could have been higher

Comments on the Quality of English Language

Good

Author Response

Dear reviewer,

Thank you very much for your kind review of our work.

We fully agree that the number of patients in our study is relatively small. This results out of the relative rarity of cerebral abscesses and the not standardised acquisition of DSC-PWI data in these patients.  We see this as a major limitation of our study. Nonetheless, we think that our data may have impact on diagnostic and therapeutic decisions in the management of patients with a brain abscess – even based on our small study population.

Thank you very much for your kind review.